# Enhanced Thermoelectric Performance of CoSb_3_ Thin Films by Ag and Ti Co-Doping

**DOI:** 10.3390/ma16031271

**Published:** 2023-02-02

**Authors:** Meng Wei, Hong-Li Ma, Min-Yue Nie, Ying-Zhen Li, Zhuang-Hao Zheng, Xiang-Hua Zhang, Ping Fan

**Affiliations:** 1Shenzhen Key Laboratory of Advanced Thin Films and Applications, Key Laboratory of Optoelectronic Devices and Systems of Ministry of Education and Guangdong Province, College of Physics and Optoelectronic Engineering, Shenzhen University, Shenzhen 518060, China; 2Univ Rennes, CNRS, ISCR (Institut des Sciences Chimiques de Rennes), UMR6226, F-35000 Rennes, France; 3BASIS International School Park Lane Harbour, Huizhou 516000, China

**Keywords:** CoSb_3_, thin film, co-doping, thermoelectric properties

## Abstract

The Skutterudites CoSb_3_ material has been the focus of research for the conversion applications of waste heat to electricity due to its ability to accommodate a large variety of ions in the cages that have been proven effective in improving the thermoelectric performance. Although the co-doped CoSb_3_ bulk materials have attracted increasing attention and have been widely studied, co-doped CoSb_3_ thin films have been rarely reported. In this work, Ag and Ti were co-doped into CoSb_3_ thin films via a facile in situ growth method, and the influence of doping content in the thermoelectric properties was investigated. The results show that all the Ag and Ti co-doped CoSb_3_ thin films contain a pure well-crystallized CoSb_3_ phase. Compared to the un-doped thin film, the co-doped samples show simultaneous increase in the Seebeck coefficient and the electrical conductivity, leading to a distinctly enhanced power factor. The high power factor value can reach ~0.31 mWm^−1^K^−2^ at 623 K after appropriate co-doping, which is two times the value of the un-doped thin film we have been obtained. All the results show that the co-doping is efficient in optimizing the performance of the CoSb_3_ thin films; the key point is to control the doping element content so as to obtain high thermoelectric properties.

## 1. Introduction

Thermoelectric (TE) power generators, which can be capable of direct and reversible conversion between heat and electricity, have a many advantages such as non-pollution, no moving parts, low maintenance, high reliability, no noise, and a long service life [1,2,3,4]. The TE devices comprise *p*-type and *n*-type materials. The performance of TE materials can be assessed by the non-dimensionless figure-of-merit *ZT*, which contains electrical conductivity *σ*, Seebeck coefficient *S* and thermal conductivity *κ* [5,6,7]. Skutterudites CoSb_3_ has been the focus investigation of conversion applications for waste heat to electricity, due to its ability to accommodate a large variety of ions in the cages, including lanthanide, alkaline earth, and alkali metals, which have been proven effective in improving thermoelectric performance [8,9,10]. These filling materials perform two tasks. One is donating electrons into the CoSb_3_ framework and increasing the electrical conductivity. The other is that filling atoms are loosely bound in the cages, leading to Einstein-like vibrational modes, which scatter more phonons and significantly reduce the thermal conductivity [11,12]. The vibrational frequencies of the filler materials are specific for their properties and tend to be different depending on their chemical groups in the periodic table. The electronic conduction is confined to the CoSb_3_ framework, electrons and phonons are scattered by the non-periodic filling [13,14], thus leading to the improvement of *ZT* value.

Recently, it has been shown that a higher *ZT* value can be realized for the filler atoms from different chemical groups that are co-doped into the skutterudite structure than that of the single-filled materials. The reason is that the electrical and thermal transports in multiple-doped skutterudites can be optimized in a relatively independent way [15,16,17,18]. The key factors here for these high *ZT* values are as follows: (1) The large atomic mass of the filler atoms leads to low frequency modes, favorable for reducing the thermal conductivity, and the introduction of structural disorder enhances phonon scattering and lowers the lattice thermal conductivity [19,20,21]; (2) Some doped atoms substitute at the Sb-sites or the Co-sites that influence the electronic structure and introduce defects, all those in favor of high electronic conductivity and low lattice thermal conductivity [22,23,24]. For instance, Shi et al. [25] prepared Ba, La, and Yb multi-filled CoSb_3_ bulk materials and obtained very high thermoelectric figure of merit *ZT* = 1.7 at 850 K. The optimum carrier density can be obtained by adjusting the total filling fraction of fillers with different charge states to reach the optimum carrier density for obtaining high power factors. In order to receive high power factors, they adjusted the total filling fraction of fillers with different charge states to obtain the optimum carrier density. In the meantime, lattice thermal conductivity can also be significantly reduced, to values near the glass limit of these materials through combining filler species of different rattling frequencies to achieve broad-frequency phonon scattering. As a result, in order to obtain a significantly improved *ZT* value, partially filled skutterudites with multiple fillers of different chemical nature allow acquiring unique structural characteristics for optimizing electrical and thermal transports in a relatively independent way. Lamberton et al. [26] studied how the charge substitution of Ge for Sb in the Yb-filled CoSb_3_ could improve the effective mass and the Seebeck coefficients at room temperature, while the lattice thermal conductivity was reduced. Rogl et al. [27] researched the structural and transport properties of *p*-type skutterudites with the nominal composition DD_0.7_Fe_2.7_Co_1.3_Sb_11.7_{Ge/Sn}_0.3_ (DD, didymium). Because of additionally introduced defects by the Fe/Co substitution and didymium filling, an increased electrical resistivity was compensated by a significantly lower thermal conductivity, and ultimately led to a high figure of merit *ZT* = 1.45 at 850 K. The co-doped CoSb_3_ bulk materials have attracted significant attention and have been increasingly studied in recent years, yet there are few reports on co-doped CoSb_3_ thin films. Sarath Kumar et al. [28] prepared In and Yb co-doped CoSb_3_ thin films by pulsed laser deposition and obtained the maximum power factor of ~0.68 Wm^−1^K^−1^ at 660 K, which was nearly four times smaller compared to that reported bulk material. On the whole, both results show that the performance of the co-doped films was still not as good as the bulk materials. Therefore, further research for preparing high-performance co-doped CoSb_3_ thin films should be conducted.

In this work, to realize this goal, detailed studies on single Ag and Ag/Ti co-doped CoSb_3_ thin films has been carried out. The doped thin film samples were prepared by Ag co-sputtering and the Ti prefabricated layer doping method. The influence of the doping amount on thermoelectric properties was explored. The results of microstructure characterization shows that all the Ag/Ti co-doped CoSb_3_ thin films contain a well-crystallized pure CoSb_3_ phase. Compared to the un-doped thin film, the co-doped samples obtained increased Seebeck coefficient and electrical conductivity. Meanwhile, the thin films with improved thermoelectric performance were obtained after appropriate co-doping and the power factor values were over 0.30 mWm^−1^K^−2^, which is twice as much for the un-doped thin film. Thus, it was revealed that the Ag/Ti co-doping can optimize the performance of the CoSb_3_ thin films.

## 2. Experimental Section

Ti target (99.99%) was fixed in the magnetron sputtering facility. The glass substrates were used and cleaned in an ultrasonic bath for 10 min in acetone, 10 min in absolute ethyl alcohol and 10 min in deionized water. The vacuum chamber was firstly pumped down to 8.0 × 10^−4^ Pa and the sputtering pressure was kept at 0.4 Pa with argon flow of 40 sccm. The Ti layer was deposited onto three different substrates with a deposition rate of about 50 nm/min and a deposition time of 5, 10 and 15 s, respectively. Then, all the samples with the precursor Ti layer were used as co-deposition substrates. Ag target (99.99%) and CoSb_3_ (99.95%) alloy targets were used for co-deposition by using the power of 1 W and 50 W, respectively. Before the film co-deposition, the above-prepared substrates were heated to 523 K. Then, the Ag-doped CoSb_3_ thin films were deposited onto the substrates containing Ti precursor layers for 2 s (direct current (DC) magnetron sputtering for Ag) and 15 min (radio frequency (RF) magnetron sputtering for CoSb_3_).

The crystal structure was determined by X-ray diffraction (XRD) (D/max2500, Rigaku Corporation, Japan) at a scanning rate of 5°/min with the 2*θ* angle range of 10–80°, using Cu Kα radiation (*λ* = 0.15406 nm) under the operation conditions of 40 kV and 40 mA. Raman spectrometer was used (RENISHAW inVia, Gloucestershire, UK), the monitor light with a wavelength of 532 nm, the power of the outgoing laser attenuated to 5%, and the spectral resolution could reach 1 cm^−1^. The surface morphology was characterized by using scanning electron microscopy (SEM, Supra 55, Zeiss, Oberkochen, Germany), while the composition analysis was performed by energy-dispersive spectroscopy (EDS, Bruker Quantax 200, Bruker, Karlsruhe, Germany). The transmittance was tested by Ultraviolet–visible spectroscopy (UV3600, Shimadzu, Tokyo, Japan), the test range is 600–2800 nm, and the wavelength interval is 2 nm. The temperature dependence of the Seebeck coefficient *S* and the electrical conductivity *σ* were simultaneously measured at the temperature range from 298 K to 623 K by using an appropriate system (SBA458, Nezsch, Bavaria, Germany). The room-temperature carrier concentration (*n*) and mobility (*μ*) were investigated by a Hall-effects measurement system (HL5500PC, Nanometrics, California, America). According to the equipment specifications, the typical measurement precisions are the following: *S* is 7%, *σ* is 7%, while *n* and *μ* are 10%.

## 3. Results and Discussion

The microstructure has an important effect on the thermoelectric properties of thin films, so that the microstructures of CoSb_3_ thin films are investigated. Figure 1 shows the XRD patterns of Ag/Ti co-doped CoSb_3_ thin films, including the un-doped and single Ag-doped thin films. Three major diffraction peaks located at ~31°, ~37° and ~44° are observed from all the patterns and they are indexed as the reflection peaks from the (013), (123) and (420) planes of CoSb_3_. Other weak peaks from the patterns can also be indexed, the results show that all of them belong to the CoSb_3_ diffraction planes, and no impurity phase can be observed within the detection limit of the XRD analysis, demonstrating that all the samples have a single CoSb_3_ phase with the body-centered cubic structure with space group Im3. The unit cell consists of 32 atoms, in which Co atoms form eight sub-cubes, forming two icosahedra voids [29,30]. As shown in Figure 1, the XRD peaks of the doped samples are progressively shifted to a small angle, which is consistent with the previous study of Zheng et al. [30]. This suggests that the crystal cell expansion, due to the doped elements entered into the lattice and Ag atoms, tends to fill the voids of the lattice [8,9,30]. In addition, the angle shift increases with the increasing Ti content, indicating that Ti atoms tend to fill the interstitial sites, similar to the behavior of Ag. It is worth noting that the peak intensities of a (013) plane for all doped samples are much higher than those of the un-doped sample, indicating that the doped films have much smaller full width at half maximum. According to the Debye–Sheerer Equation, a smaller full width at half maximum means a larger crystal size. Additionally, the volume fraction of the peak corresponding to a (013) plane is over 50% of the whole pattern for all the co-doped thin films, suggesting a preferred orientation growth to this direction. Generally, the thin films will have a larger crystal size when there is a tendency of preferred growth to a certain direction, and this is always favorable for the enhancement of electronic and thermoelectric transport.

Figure 2 shows the Raman test results, which range from 80–230 cm^−1^. There are four vibration modes of Raman phonons corresponding to F_g2_, E_g1_, A_g1_ and E_g2_ that can be observed at the peaks of 106 cm^−1^, 136 cm^−1^, 149 cm^−1^ and 182 cm^−1^, respectively. The peak of the Raman spectrum corresponds to the theoretical calculated peak of CoSb_3_ [3], and no other peaks are observed. It is further confirmed that the thin film samples have a single-cubic crystal structure. It is worth noting that the intensity and full width at half the maximum diffraction of the peaks do not change significantly before and after Ag or Ag/Ti doping. This phenomenon shows that the atoms in the crystal structure have stable bonding states. However, with the increase in Ti doping, the Raman peak also produces a slight shift, which may be caused by the change of lattice constant combined with the XRD test results.

Figure 3a–e displays the surface morphologies for the un-doped, single Ag-doped and Ag/Ti co-doped thin films, respectively. Table 1 shows the element contents and thickness of the thin films. It can be observed that the thickness of the un-doped thin film is ~170 nm. The Ag has a stable content of around 0.2 at.% in all the co-doped thin films whose thickness can reach ~210 nm. In addition, the Ti content of the co-doped sample is about 0.1 at.%, 0.4 at.% and 0.6 at.%, respectively, and the thickness of the thin films increases with the increased Ti content; the maximum value is ~230 nm. The SEM results reveal that all the thin films exhibit a smooth and dense surface across all regions. Few changes can be observed after Ti doping, and no cluster or porous defects can be found in the grain/particle boundary. These characteristics indicate that these thin films exhibit few defects that are favorable for electron transport. Thus, by combing the XRD results, it can be concluded that all the co-doped thin films are well crystallized. Figure 3f–f4 displays the EDS mapping for the thin films with the largest Ti contents, which indicates that all the elements are homogenously distributed in the thin films for the co-doped samples.

Figure 4 shows the relationship between (*αhν*)^2^ and *hν*. Value x of the intersection of the tangent line and the horizontal coordinate is the optical band gap of the thin film sample. There may be some error between the optical band gap calculated by the transmittance measurement of Ultraviolet–visible spectroscopy (UV-VIS) spectrum and the energy difference between the conduction band bottom and the valence band top (band gap width) calculated theoretically, but there should be a corresponding relationship in principle. The impurity level formed by the added impurities is located in the gap, the Fermi level position will change, and the measured optical band gap (impurity semiconductor) generally will be bigger than the band gap of the material, but it can also reflect the changing trend of the band gap of the thin film. It can be seen in Figure 4 that the *hν* (Eg), which is the x value of the intersection of the horizontal coordinate of the film sample of undoped CoSb_3,_ is ~1.53 eV, the Eg of the film sample for the CoSb_3_ with single Ag doping is ~1.49 eV, and the Eg of the film samples of Ag/Ti-doped CoSb_3_ is ~1.45–1.48 eV. In consequence, the Eg of the doped thin films is smaller than that of the un-doped thin film. With the slight increase in the doping amount, the optical band gap is slightly reduced. The Eg of the Ag/Ti_15s_ thin film reaches the minimum value. In terms of electron transport, the position of the Fermi energy level will be changed. The Fermi position moves towards the direction of the conduction band and enters into the conduction band, which is beneficial to the transport of electrons, making the material appear as an *n*-type semiconductor, which is also consistent with the Seebeck coefficient and carrier concentration results. 

Figure 5 shows the thermoelectric properties of the Ag/Ti co-doped thin films. For comparison, the thermoelectric properties of the un-doped and the Ag-doped samples are also added in the figures. From Figure 5a, the Seebeck coefficient *S* displays a negative sign over the entire temperature range, indicating the *n*-type behavior. The absolute *S* value is about 25 μVK^−1^ for the un-doped sample and about 45 μVK^−1^ for the single Ag-doped thin film at room temperature. After Ti co-doping, the room-temperature absolute *S* slightly changed, and the value for the Ag/Ti co-doped thin films was higher than that of the un-doped sample, but was lower than that of the single Ag-doped thin film at high temperature. The maximum absolute *S* value of the Ag/Ti co-doped thin films is, respectively, 206 μVK^−1^, 229 μVK^−1^, and 183 μVK^−1^ for the different doping content of Ti. The carrier concentration *n* and mobility *μ* of the CoSb_3_ thin films are shown in Figure 5b. As can be seen from the figure, the negative carrier concentration demonstrates that all the thin films are *n*-type semiconductor thermoelectric materials. The absolute *n* values of Ag-doped and slight Ag/Ti co-doped thin films are lower than the un-doped thin film, and decrease with increasing content. This indicates that the addition of a small amount of elements reduces the carrier concentration. As can be seen in Figure 5b, the absolute *n* value of the Ag/Ti_15s_ thin film reaches the maximum value, indicating that a large number of filler atoms increase the carrier concentration. Compared to the carrier concentration, the trend in mobility is the opposite. When doping slightly, the mobility of the thin film increases with the increase in the doping amount. Additionally, the mobility of the *μ* value of the Ag/Ti_5s_ thin film is higher than other thin films. In addition, the increase rate of mobility is greater than the decrease rate of carrier concentration. Electrical conductivity *σ* is calculated by the formula *σ* = *nμe*, where *e* is the electron charge. Therefore, the electrical conductivity increases with the slight increase in the doping amount, as shown in Figure 5c. Combined with the carrier concentration, electrical conductivity *σ* of the Ag/Ti_5s_ thin film is higher than other thin films’, except for 623 K. In Figure 5c, *σ* of the un-doped sample is 0.16 × 10^4^ Sm^−1^ at room-temperature and it increases with the increasing temperature, suggesting its semiconductor characteristics. Similar result can be observed for the single Ag-doped thin film. Electrical conductivity *σ* obviously increases after the Ti co-doping and shows the same trend with the temperature as the un-doped sample. Comparatively, co-doped samples show high electrical conductivity at high temperatures, and the maximum *σ* values of 0.68 × 10^4^ Sm^−1^ at 623 K are obtained from the Ag/Ti_15s_. From the results it can be concluded that the Ag is more beneficial for improving the *S*, and the Ti doping is more efficient for enhancing the *σ*. The power factor *PF*, which is calculated by the formula *PF = S^2^σ*, is depicted in Figure 5d. Compared to the un-doped thin film, the *PF* value is obviously enhanced after Ag and Ti co-doping due to the simultaneous increase in both the Seebeck coefficient and the electrical conductivity. The *PF* value for the co-doped thin films increases firstly with the increasing Ti content, then decreases when Ti content further increases. Although the *PF* value of the co-doped thin films is higher than that of the single Ag-doped sample in the testing temperature range from 298 K to 478 K, as shown in Figure 5d, it becomes smaller when the temperature is over 500 K. The maximum *PF* value is about 0.31 mWm^−1^K^−2^ at 623 K for Ag/Ti_10s_, which is two times that of the *PF* value for the un-doped sample.

## 4. Conclusions

In this study, detailed studies on magnetron sputtering deposition for the preparation of Ag/Ti co-doped CoSb_3_ thin films have been performed. The importance of the co-doping element type and content on the properties of CoSb_3_ thin films has been demonstrated. The results show that all the Ag/Ti co-doped CoSb_3_ thin films contain a well-crystallized pure CoSb_3_ phase. With the increase in the Ag/Ti doping amount, the optical bandgap decreases gradually. Compared to the un-doped thin film, the co-doped samples show simultaneous increase in the Seebeck coefficient and the electrical conductivity, leading to a distinctly enhanced power factor. As expected, thin films with improved thermoelectric performance have been obtained after appropriate co-doping and the highest power factor value over 0.30 mWm^−1^K^−2^ at 623K for Ag/Ti_10s_, which is twice as big for the un-doped thin film. Thus, all the results show that the Ag/Ti co-doping is efficient in optimizing the performance of the CoSb_3_ films.

## Figures and Tables

**Figure 1 materials-16-01271-f001:**
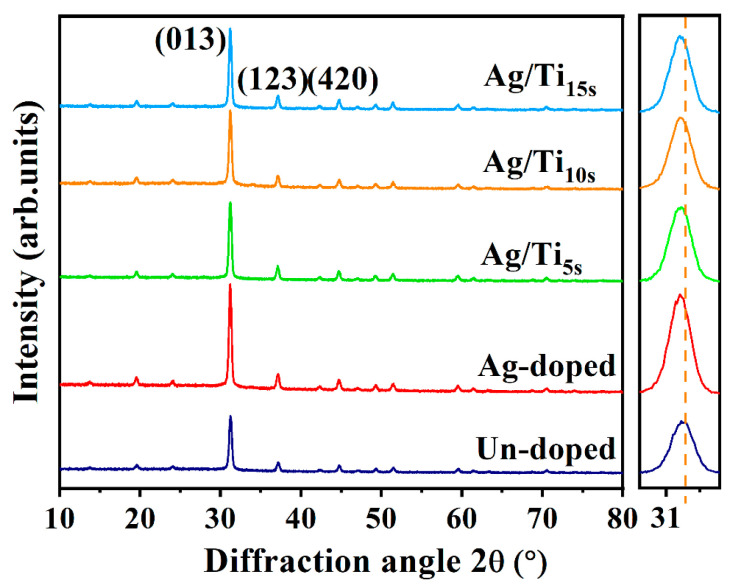
The XRD patterns for Ag/Ti co-doped CoSb_3_, the un-doped and single Ag-doped thin films.

**Figure 2 materials-16-01271-f002:**
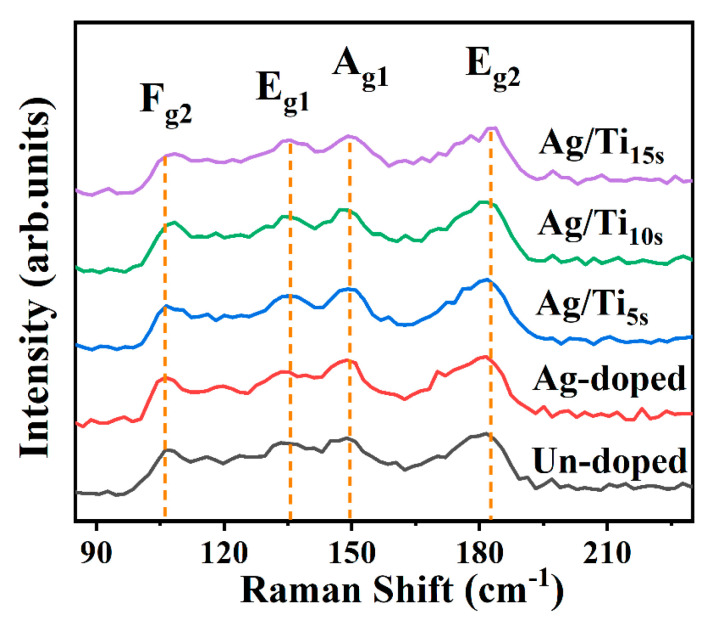
The Raman patterns for Ag/Ti co-doped CoSb_3_, the un-doped and single Ag-doped thin films.

**Figure 3 materials-16-01271-f003:**
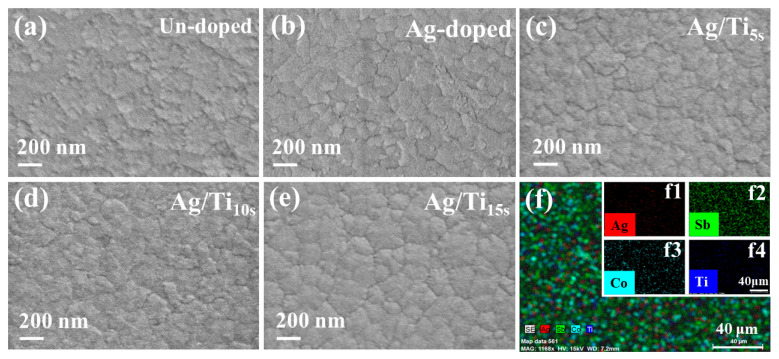
(**a**–**e**) Surface morphology images of the un-doped, single Ag-doped and the Ag/Ti co-doped thin films and (**f**–**f4**) EDS mappings.

**Figure 4 materials-16-01271-f004:**
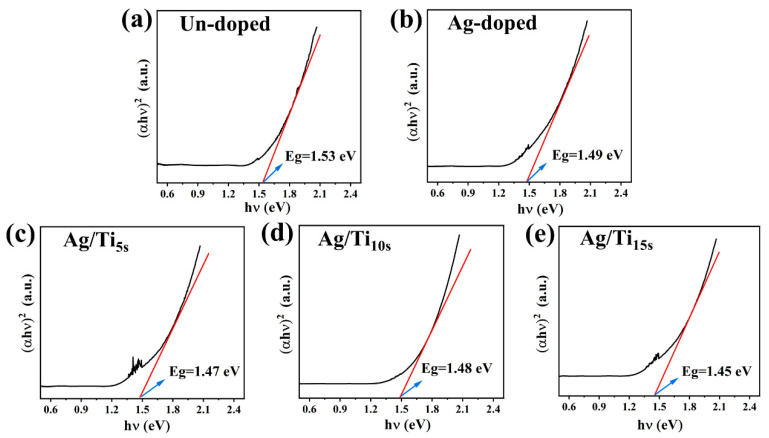
(**a**–**e**) The relationship between (*αhν*)^2^ and *hν* for Ag/Ti co-doped CoSb_3_, the un-doped and single Ag-doped thin films.

**Figure 5 materials-16-01271-f005:**
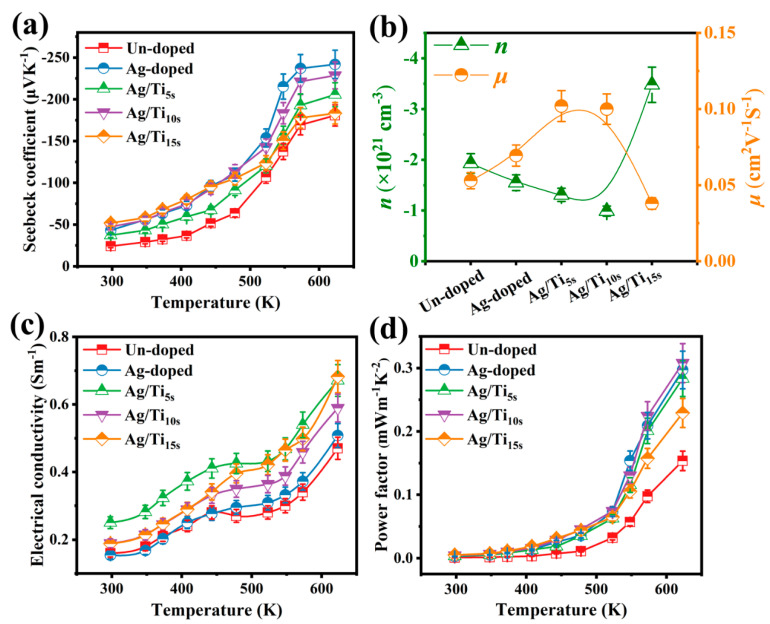
Thermoelectric properties of all the thin films: (**a**) Seebeck coefficient; (**b**) Carrier concentration and mobility; (**c**) Electrical conductivity; (**d**) Power factor from 273 K to 623 K.

**Table 1 materials-16-01271-t001:** The element content and thickness of the thin films.

Sample	Ag (at.%)	Ti (at.%)	Thickness (nm)
Un-doped	--	--	170 ± 5
Ag-doped	~0.2	--	210 ± 5
Ag/Ti_5s_	0.3	0.1	223 ± 5
Ag/Ti_10s_	0.1	0.4	225 ± 5
Ag/Ti_15s_	0.2	0.6	230 ± 5

## Data Availability

Data will be made available on request.

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
