# Peer review of "Enhanced Thermoelectric Performance of CoSb3 Thin Films by Ag and Ti Co-Doping"

_materials, 2023, doi:10.3390/ma16031271_

Round 1

Reviewer 1 Report

The skutterudite CoSb3 has emerged as one of the most studied candidate materials for thermoelectric applications. However, TE properties of CoSb3 thin films are lower than those of bulk materials. But interesting work is going-on to improve the TE feature on thin films.    The present study is well presented in a coordinated manner. I have some queries about the present manuscript as:

 1. Similar studies of doping on CoSb3 thin films with different impurities are reported by many authors including Dr. P. Fang- the communicating authors. A comparison with literature will enhance importance of the present study.

2. Figure 1. In the manuscript is very similar to the XRD patterns of un-doped and Ag doped CoSb3 thin films given in authors earlier paper _ Inorg. Chem. Front., 2018, DOI: 10.1039/C8QI00207J. The distinguishable features should be discussed with reference to the  paper.

3. The optical bandgap decreases by doping skutterudite thin films. How co-doping with Ti is further helpful?

Reviewer 2 Report

In the manuscript "Enhanced thermoelectric performance of CoSb3 thin films by Ag and Ti co-doping" skutterudites CoSb3 material has been the focus of research for waste heat to electricity conversion applications due to its ability to accommodate a large variety of ions into the cages which have been proven effective in improving the thermoelectric performance. The paper and results is interesting for scientific community.
The comments on this study are as follows:
1. On page 3 in the Results and Discussion part in the sentence "Other weak peaks from the patterns can also be indexed,
the results show that all of them belonging to the CoSb3 diffraction planes ..." it is better to change "XRD spectrometer" with a "XRD analysis".
2. On page 4 in the second sentence should be exclude "diffraction peak" and leave only the peak.
3. On page 3 "Besides, the angle shift increases with increasing Ti content, indicating Ti is more likely filling into the void of the lattice, similarly to the behavior of Ag". What voids are we talking about, what solid solutions are formed, and what positions do the alloying elements occupy? More detailed description needed.
4. It is necessary to give the space group and/or the structural type of the phase CoSb3.
5. In the description of the XRD analysis the possible texturing of the sample is indicated in this regard it is necessary to indicate how the properties were measured relative to the texture.
6. It is better to increase the contrast of figure 3.

Reviewer 3 Report

In this work, detailed studies on magnetron sputtering deposition for preparing Ag/Ti co-doped CoSb3 thin films have been performed. The importance of the co-doping element type and content on the properties of CoSb3 thin films has been demonstrated. For future it would be interesting to study zT value for such system, as even PF will be high, high values of thermal conductivity might decrease overall figure-of-merit, thus leading to losing interest to the TE field.

Round 2

Reviewer 1 Report

The revised presentation is satisfactory.